# The Residual Efficacy of SumiShield™ 50WG and K-Othrine^®^ WG250 IRS Formulations Applied to Different Building Materials against *Anopheles* and *Aedes* Mosquitoes

**DOI:** 10.3390/insects13020112

**Published:** 2022-01-20

**Authors:** Rosemary Susan Lees, Giorgio Praulins, Natalie Lissenden, Andy South, Jessica Carson, Faye Brown, John Lucas, David Malone

**Affiliations:** 1Vector Biology Department, Liverpool School of Tropical Medicine, Pembroke Place, Liverpool L3 5QA, UK; giorgio.praulins@lstmed.ac.uk (G.P.); Natalie.Lissenden@lstmed.ac.uk (N.L.); Andy.South@lstmed.ac.uk (A.S.); jessica.carson@lstmed.ac.uk (J.C.); Faye.Brown@liverpool.ac.uk (F.B.); 2Liverpool Insect Testing Establishment (LITE), Liverpool School of Tropical Medicine, 1 Daulby Street, Liverpool L7 8XZ, UK; 3Institute of Infection, Veterinary and Ecological Sciences, Department of Livestock and One Health, The University of Liverpool, Liverpool L69 3BX, UK; 4Environmental Health Division, Sumitomo Chemical (UK) plc, 200 Shepherds Bush Rd, London W6 7NL, UK; johnlucas@vectorcontrolworks.net; 5Innovative Vector Control Consortium (IVCC), Liverpool School of Tropical Medicine, Liverpool L3 5QA, UK; David.Malone@gatesfoundation.org; 6Bill & Melinda Gates Foundation, 500 5th Ave N, Seattle, WA 98109, USA

**Keywords:** indoor residual spray (IRS), vector control, *Anopheles*, *Aedes* *aegypti*, *Culex quinquefasciatus*, clothianidin, neonicotinoids, pyrethroid, deltamethrin, insecticide resistance, Sumishield, K-Othrine

## Abstract

**Simple Summary:**

The *Anopheles* mosquitoes that transmit malaria are targeted by the use of indoor residual sprays (IRSs), insecticides applied to the walls of homes to kill mosquitoes that rest there when coming into houses in search of a blood meal. K-Othrine^®^ is an IRS based on the pyrethroid deltamethrin and is widely used against mosquitoes that transmit malaria. SumiShield™ 50WG is an IRS based on the insecticide clothianidin, developed to kill mosquitoes that have become resistant to other forms of insecticide. These products were applied to cement, wood, and mud tiles, representative of typical building materials in areas where malaria is endemic. For 18 months, the ability of these treated surfaces to kill adult female mosquitoes exposed to them was measured. The clothianidin IRS was highly effective against insecticide susceptible and resistant strains of *Anopheles gambiae* and *An. funestus*, key malaria vector species, with an improved performance compared to deltamethrin IRS, though was not so effective against *Aedes aegypti* or *Culex quinquefasciatus*. Both IRS formulations were shown to be more effective and long-lasting on cement and mud than on wood tiles.

**Abstract:**

Insecticides with novel modes of action are required to complement the pyrethroids currently relied upon for controlling malaria vectors. One example of this is the neonicotinoid clothianidin, the active ingredient in the indoor residual spray (IRS) SumiShield™ 50WG. In a preliminary experiment, the mortality of insecticide-susceptible and resistant *An. gambiae* adults exposed to filter papers treated with this IRS product reached 80% by 3 days post-exposure and 100% by 6 days post-exposure. Next, cement, wood, and mud tiles were treated with the clothianidin or a deltamethrin-based IRS formulation (K-Othrine WG250). Insecticide resistant and susceptible *Anopheles* and *Aedes* were exposed to these surfaces periodically for up to 18 months. Pyrethroid resistant *Cx. quinquefasciatus* was also exposed at 9 months. Between exposures, tiles were stored in heat and relative humidity conditions reflecting those found in the field. On these surfaces, the clothianidin IRS was effective at killing both susceptible and resistant *An. gambiae* for 18 months post-treatment, while mortality amongst the resistant strains when exposed to the deltamethrin IRS was not above that of the negative control. Greater efficacy of clothianidin was also demonstrated against insecticide resistant strains of *An. funestus* compared to deltamethrin, though the potency was lower when compared with *An. gambiae*. In general, higher efficacy of the clothianidin IRS was observed on cement and mud compared to wood, though it demonstrated poorer residual activity against *Ae.aegypti* and *Cx. quinquefasciatus*.

## 1. Introduction

Insecticide-treated nets (ITNs) and indoor residual spraying (IRS) continue to be the two primary methods used in vector control strategies against malaria [1]. Insecticides are key to the control of other mosquito-borne diseases, such as arbovirus infections transmitted by *Aedes*, as well as the control of nuisance biters, such as *Culex* species. IRS can greatly reduce disease transmission risk by decreasing the survival of mosquitoes as well as reducing biting intensity [2]. However, the substantial progress made in the reduction of disease transmission, particularly malaria, is under threat from the increasing spread of insecticide resistance to conventional insecticides, namely pyrethroids, carbamates, and organophosphates [3,4]. Although there are other vector control tools (e.g., larval source management) and new technologies in development (e.g., transgenic mosquitoes and the use of symbionts), insecticides remain essential in the control of endophilic vectors. Therefore, there is an urgent need to develop new insecticides and formulations for IRS, effective against mosquitoes that exhibit resistance to currently approved insecticide classes. To investigate the risk of cross resistance in pyrethroid resistant populations, it is valuable to test new chemistries against well-characterised strains of insecticide susceptible and resistant mosquitoes. Laboratory strains can be maintained in a controlled and consistent manner to allow comparisons to be made between studies and between compounds [5]. 

Currently, there are five insecticide classes used in IRS products prequalified by the World Health Organization (WHO): pyrethroids, carbamates, organophosphates, and the most recently added, neonicotinoids [6]. Clothianidin is a neonicotinoid, a class of insecticides that act as agonists of nicotinic acetylcholine receptors within mosquitoes. This novel mode of action gives neonicotinoids the potential to provide control of vectors in areas of high pyrethroid resistance. An IRS formulation containing clothianidin, SumiShield™ 50WG, has been shown to be effective in Phase I trials [7,8], in Phase II trials in areas of high-intensity insecticide resistance [9,10], and in Phase III trials in India [11] and Tanzania [12]. 

The success of an IRS program depends on several factors, including vector resting behaviour, residual efficacy of insecticides, spray coverage, and the quality of spraying [13]. The typical target residual efficacy of an IRS product is 6 months, but this efficacy can vary greatly depending on the nature of the sprayed surfaces [14,15,16]. A laboratory study using a deltamethrin IRS (K-Othrine WP 5%) against susceptible *Anopheles stephensi* Liston found that it retained efficacy (defined by the WHO as >80% mortality) for 2 months on mud, 4 months on plaster and wood, and 4.5 months on cement [14]. Similarly, a study in Cameroon showed that deltamethrin IRS (K-Othrine^®^ WG 250) sprayed on concrete walls had the longest residual efficacy (6 months), followed by mud (4.5 months), then wood (3.5 months) [16]. A study in Zanzibar determined that a pirimiphos-methyl based IRS (Actellic^®^ 300CS) applied on various wall surfaces (mud, oil, water painted, lime washed walls, un-plastered cement, and stone blocks) remained effective for at least 8 months after spraying [17]. 

Some studies have investigated the residual efficacy of SumiShield™ 50WG on different surfaces, including mud and cement [9,11]. In India, against insecticide resistant *An. culicifacies*, Sreehari et al. [9] observed the same formulation to have a residual life of 15–25 weeks on cement and mud-plastered houses, depending on the mosquito holding period post-exposure (24-h mortality to 120-h mortality). An additional 32% of mosquitoes died between 24 and 120 h when exposed on a treated cement wall, and 40% on a mud wall, leading to clothianidin being referred to as causing additional kill activity over time in addition to that observed using standard protocols. Similarly, Uragayala et al. observed that the residual efficacy of SumiShield™ 50WG in houses against *An. culicifacies* increased from 5 to 6 months when the holding period was extended from 24 to 120 h [11]. 

Residual efficacy experiments of IRS formulations applied to different surface types in a controlled laboratory environment can provide vital information to help make predictions about efficacy in operational use in different settings. Temperature and humidity can be controlled while the surfaces are treated and stored, to standardise conditions throughout a long experiment and between treatments. Spray application can also be performed accurately, using a Potter Tower [18,19]. The Potter Tower is recommended by the WHO for laboratory studies to test insecticide residual activity and is an internationally recognised method of chemical spraying [20]. Other studies have assessed the residual efficacy of SumiShield™ 50WG in a field setting, using a compression sprayer to treat huts. However, they often show high variability in spray uniformity, as illustrated by a Phase III study in India, where the target dose was 300 mg AI/m^2^, but the mean dose applied was 516.6 mg AI/m^2^, with some villages having an actual to target dose ratio as high as x2.4 [11]. A similar study in Tanzania achieved a closer target dose (average 363.4 mg AI/m^2^) [12], making comparisons between studies difficult. Kweka et al. [12] and Uragayala et al. [11] used mud-only versus mud-plastered walls with lime coating, with mortality based on a 168 and 120 h holding period of mosquitos post-exposure, respectively. The residual efficacy of IRS formulations on mud surfaces may be affected by the specific physical and chemical properties of the mud.

The controlled laboratory experiment reported here aimed to assess the residuality of SumiShield™ 50WG (hereafter referred to as ‘clothianidin IRS’). Testing was conducted on surfaces commonly used for building houses in areas where IRS is employed (i.e., mud, cement, and plywood). K-Othrine^®^ 250WDG (hereafter referred to as ‘deltamethrin IRS’), a pyrethroid-based product that has been widely employed for IRS, was included as a comparator. First, the speed of kill against a pyrethroid susceptible and resistant strain of *An. gambiae* was tested in a WHO tube assay to determine the most appropriate holding period. Then, pyrethroid-resistant and -susceptible laboratory strains of *An. gambiae*, *An. funestus*, *Ae. aegypti*, and *Cx. quinquefasciatus* were exposed to treated mud, cement, and wood surfaces. Finally, the effect of increasing the exposure time was investigated in susceptible and resistant laboratory strains of *Ae. aegypti*, and susceptible *An. gambiae*.

## 2. Materials and Methods

### 2.1. Mosquito Strains

All mosquitoes were reared from colonies maintained in the Liverpool Testing Establishment (LITE) at the Liverpool School of Tropical Medicine (LSTM), Liverpool, UK, according to the methods described by [5]. Adult female mosquitoes, 2–5 days old, allowed to mate but not blood feed, were used for all bioassays. Seven mosquito strains were used in the residuality experiment. *An. gambiae* s.l VK7 2014 (highly resistant to pyrethroids and DDT through a combination of target site and metabolic resistance) and Kisumu (susceptible), *An. funestus* FuMoz-R (moderately resistant to pyrethroids and DDT through metabolic resistance), and Fang (susceptible), *Ae. aegypti* Cayman (highly resistant to pyrethroids, DDT and carbamates due to target site resistance), and New Orleans (susceptible). Resistance profiles of these strains are available in Williams et al. [5]. A strain of *Cx. quinquefasciatus* (Muheza) was also tested. Colonised from coastal Tanzania in the early 1990s and since selected for permethrin resistance [21], this strain is highly resistant to permethrin, deltamethrin, DDT, and dieldrin, and susceptible to fenitrothion and propoxur (authors’ unpublished data).

### 2.2. Test Surfaces

Three surfaces, representative of materials that may be used to construct dwellings in areas of IRS application, were treated for efficacy testing: wood, cement, and mud.

The wood surfaces (12 cm^2^ squares) were cut from untreated beechwood approximately 1 cm thick.

The cement surfaces (10 cm diameter circles, ~5 mm thick) were prepared by sieving sand and cement powder separately to remove any dirt and large particles then combining ~600 mL of sand, 600 mL of cement powder, and 300 mL of purified (Millipore) water, mixing thoroughly to a thick paste. Petri dishes 10 cm in diameter were filled, pushing the surface down firmly to ensure there were no gaps and flattening until the top surfaces were smooth. Surfaces were dried at 27 ± 2 °C and 80 ± 10% RH) for a minimum of 30 days prior to testing.

The mud surfaces (10 cm diameter circles) were made from unfired mud bricks provided by the Institut de Recherche en Sciences de la Sante (IRSS), Burkina Faso. The mud was collected from their field station in Vallée du Kou 7, Burkina Faso (4°24′ W, 11°24′ N). Mud bricks were broken down into dust, reconstituted by adding small amounts of purified (Millipore) water, and mixing using hands, continuing to add water until the mud was firm and smooth in consistency. This mud was used to fill metal molds that hold a cylinder of mud, approximately 1 cm deep and with a diameter of 10 cm. Mud surfaces were stored in a climate-controlled stability cabinet (27 ± 2 °C and 80 ± 10% RH) for a minimum of 30 days to allow them to dry and produce a smooth mud surface. A mud sample was also supplied to ACS Testing Ltd. (Poole, Dorset) to determine the physical characteristics and chemical properties of the mud (Appendix A, Appendix A).

### 2.3. Preparation of Test Surfaces

SumiShield^TM^ 50WG (clothianidin indoor residual spray (IRS), Sumitomo Chemicals Ltd., London, UK) and K-Othrine^®^ WG250 (deltamethrin IRS, Bayer Environmental Science) formulations were diluted in purified (Millipore) water and applied to each test surface using a ‘Potter Tower’ (Potter Precision Laboratory Spray Tower, Burkard Scientific, Rickmansworth, UK). The target application rate was 300 mg AI/m^2^ for clothianidin and 25 mg AI/m^2^ for deltamethrin, the manufacturers’ recommended application rate. Negative control surfaces were sprayed with purified (Millipore) water only. Prior to use, the Potter Tower was calibrated to ensure less than 10% variation in spray density across the treated surface and less than 10% variation in spray weight between applications. Seven replicate plates of each treatment (clothianidin IRS, deltamethrin IRS, and control) were treated per surface type (wood, cement, mud) to produce a set of plates for bioassays (3 plates of each type and treatment) and spare reserves (4 plates of each). Two strains (resistant and susceptible) of each species were exposed to the same set of plates at each time point, a different set of plates for each species. Strains of the same species were exposed to plates on the same day, with susceptible strains being tested first for each species. The same surfaces were used for bioassays at every time point in the experiment, except where surfaces were damaged. Where this happened, the surfaces were replaced with reserve plates. After spraying and between bioassays, the surfaces were stored in a climate-controlled stability cabinet (30 ± 2 °C, 80 ± 10% RH), vertically and unsealed, with air circulation and in the dark to represent typical field conditions.

### 2.4. WHO Susceptibility Tube Bioassay

#### 2.4.1. Investigating Additional Mortality beyond 24 h Post-Exposure to Clothianidin

Because clothianidin has been seen to have additional kill activity over time, beyond the typical 24-h holding period, a preliminary test, using WHO susceptibility tube bioassays, was conducted to determine how long mosquitoes should be held after exposure to clothianidin IRS to record maximum mortality. Filter paper (Whatman No. 1) was cut into 12 cm × 15 cm pieces. In total, 264 mg of clothianidin were dissolved in 20 mL of purified (Millipore) water, and 2 mL were pipetted onto each paper to give a surface concentration of 13.2 mg AI per paper, or 733.3 mg AI/m^2^. Eight replicate papers were made, alongside six negative water-only control papers, and a positive control paper treated with 275 AI mg/m^2^ fenitrothion. Papers were dried overnight in a fume hood before being stored in silver foil at 5 °C until use. All bioassays were performed within 1 month of the papers being made.

Twenty-five mosquitoes were exposed to each clothianidin or control paper for 60 min, or fenitrothion for 120 min, in a standard WHO tube bioassay [22]. Knockdown was scored 30 and 60 min post-exposure and mortality was scored 24 h post-exposure, then daily until 7 days post-exposure.

#### 2.4.2. Investigating the Effect of Varied Exposure Time on Clothianidin Efficacy

An additional study investigated how varying the time of exposure to clothianidin IRS affected mosquito mortality. Mosquitoes were exposed to test filter papers in a WHO tube susceptibility bioassay, using the same methods for preparing papers, conducting bioassays as described above. Susceptible (New Orleans) and resistant (Cayman) *Ae. aegypti*, and susceptible *An. gambiae* (Kisumu) were exposed to clothianidin IRS treated papers for a range of exposure times (15 min to 7 h). Mosquitoes were exposed to negative water-only controls for 60 min [22]. Knock down was scored 60 min post-exposure, and mortality was scored at 24, 48, and 72 h post-exposure.

### 2.5. WHO Cone Bioassay

#### Residual Efficacy of Clothianidin IRS over Time

Each strain of mosquito was exposed to three replicate plates of each treatment and surface combination using the WHO cone bioassay [20]. Bioassays were repeated 24 h and 1, 3, 5, 7, 9, 12, and 18 months after treatment of the surfaces, with the following exceptions: at month 9, Cayman and New Orleans strains (*Ae. aegypti*) were not available and the time point was omitted, and Muheza (*Cx. quinquefasciatus*) was only tested at a single time point (9 months) using the spare reserve plates.

Ten mosquitoes were aspirated into a plastic cone plugged with cotton wool, applied to each surface, held on a board at 45 degrees, and left for 30 min before being aspirated off into a holding cup and held in a stability cabinet at 27 ± 2 °C and 70 ± 10% RH for 24 h, with access to a 10% sugar solution provided on cotton wool [20]. Mosquitoes were scored for knock down or mortality at the end of exposure (30 min), 24 h post-exposure, and daily for 7 days. A preliminary study (data not shown) found that ≥87% (*N* = 50, 5 replicate cups) of non-exposed 2–5-day old females of each strain survived for 7 days in holding cups under these holding conditions and that in most strains survival was ≥95%.

### 2.6. Data Analysis

Post-exposure knockdown and daily mortality over 7 days is reported as an average of the cone test results from three replicate surfaces or of three replicate WHO susceptibility tube tests, corrected for the control mortality using Abbott’s formula [23]. Standard error was calculated between replicates of each strain and each treatment.

## 3. Results

### 3.1. WHO Tube Bioassay: Additional Mortality beyond 24 h Post-Exposure to Clothianidin

Following exposure to clothianidin IRS for 60 min in a WHO tube assay, >99% of susceptible Kisumu and resistant VK7 2014 were killed within 7 days, and mortality reached 80% in both strains by 3 days post-exposure (Figure 1).

### 3.2. WHO Cone Test: Residual Efficacy of Clothianidin IRS over Time

Because mortality was seen to exceed 80% mortality and start to plateau 72 h after exposure to clothianidin in the preliminary experiment, all results presented for the residual efficacy assay show cumulative mosquito mortality at 72 h (3 days) post-exposure. Figures showing 24 h and 120 h mortality are available in the Appendix A (Appendix A). Both clothianidin and deltamethrin killed >90% of susceptible *An. gambiae* for 18 months after surfaces were treated, though there was some variability in the results on mud surfaces (Figure 2). Clothianidin was also very effective against resistant *An. gambiae*, which were not killed by the deltamethrin, with 100% mortality observed at 18 months (with exceptions at 3 and 9 months where mortality dropped). Mortality was less consistent over time in both resistant and susceptible *An. funestus* and varied more between surface types. However, clothianidin killed over 50% of exposed susceptible *An. funestus* over the 18 months, and consistently performed better than deltamethrin against the resistant strain.

To evaluate the additional benefit that might be achieved by a non-pyrethroid IRS used against pyrethroid-resistant *Anopheles* populations, the difference in 72-h mortality between clothianidin and deltamethrin IRS treatments was calculated for each pair of treatments (Figure 3). In *An. gambiae*, clothianidin induced very similar mortality to deltamethrin against the susceptible strain on cement and mud, and slightly worse on wood surfaces, but it killed a greater proportion of the resistant strain on all surfaces at all time points. With *An. funestus*, deltamethrin killed more of the susceptible strain in most time points, particularly on wood, but clothianidin outperformed against the resistant strain in all but two replicate tests. There is a trend towards better performance of clothianidin relative to deltamethrin in the later time points, suggestive of greater residual efficacy of this insecticide, particularly on cement and mud surfaces. In most cases, this is a result of the declining performance of deltamethrin, though the mortality in resistant *An. funestus* exposed to clothianidin is lower in months 1–7 than in months 9–18 (Figure 2).

Although SumiShield™ 50WG was specifically designed to target malaria vectors, its efficacy against *Ae. aegypti* was assessed in parallel. An opportunistic bioassay was also performed at 9 months post-treatment against the resistant Muheza strain of *Cx. quinquefasciatus*. This used backup surfaces, which were made alongside the surfaces used for bioassays but previously untested. To control for any differences between these surfaces and those which had been used before, Kisumu (susceptible *An. gambiae*) were also exposed to these surfaces. The results from the backup surfaces matched those from the results of the standard Kisumu bioassays at this time point (data not shown). Mortality in susceptible New Orleans and resistant Cayman *Ae. aegypti* strains exposed to clothianidin-treated surfaces never exceeded 50%; Deltamethrin also performed poorly against the resistant strain but killed 100% of the susceptible strain up to 18 months post-treatment, except on mud, where mortality dropped from 5 months (Figure 4A). For *Ae. aegypti*, there was no measured advantage of clothianidin over deltamethrin in any bioassays, even against the resistant strain (Figure 4B). Results were quite variable between replicate bioassays with *Cx. quinquefasciatus* (Figure 5), but overall, where an average of 60% and 40% were killed by Deltamethrin-treated cement and wood surfaces, respectively, 5% were killed by Deltamethrin-treated mud surfaces by 72 h post-exposure. Clothianidin IRS-treated wood, mud, and cement killed an average of 0, 22, and 25% of the exposed Muheza females, respectively. The observed mortality was 100% in Kisumu exposed to all deltamethrin and clothianidin IRS treated surfaces, and in both strains exposed to control tiles, mortality was <20% (data not shown).

To assess the duration of residual efficacy considering the speed of action of the IRS treatments, we plotted the average mortality over the observation time (0–168 h post-exposure) for each bioassay, marking the observation point at which 80% mortality was reached. Following exposure to clothianidin, residual efficacy (≥80% mortality by 24 h after exposure) lasted for the full 18 months of the experiment in the Kisumu (Appendix A) and VK7 2014 (Appendix A) strains of *An. gambiae*, except for month 3 when mortality was anomalously low in VK7 2014 but recovered in month 5 onwards. Against *An. funestus* (Figure 6), the mortality of the susceptible strain Fang fell below 80% at 12 months on wood and though the mortality in the resistant strain FUMOZ-R dropped below 80% for several months on wood and in month 7 on mud, it recovered in months 9, 12, and 18. *An. funestus* were killed more slowly by clothianidin than *An. gambiae*, particularly the resistant strain FUMOZ-R. The 80% threshold was never reached against either New Orleans (Appendix A) or Cayman (Appendix A) strains of *Ae. aegypti*. In contrast, deltamethrin IRS exceeded the 80% threshold for the 18 months of the study against all susceptible strains (Kisumu, Fang, and New Orleans, Appendix A), except for Fang and New Orleans on mud surfaces, which dropped below the threshold at 12 and 18 months, though the threshold was again reached against New Orleans at 18 months post-treatment. Against the resistant strains (VK7 2014, FUMOZ-R, and Cayman, Appendix A), mortality never reached 80% on any surface treated with deltamethrin IRS. Details for control surfaces are shown in the Appendix A, Appendix A.

To assess the effect of application to different surfaces on the residual efficacy of clothianidin IRS, the mortality at 72 h was aggregated for all time points by strain and surface type (Figure 7). A figure showing mortality at 120 h post-exposure is presented in the Appendix A, Appendix A. Across all mosquito strains tested, poorer efficacy of clothianidin IRS on wood surfaces was shown, except for the resistant *Ae. aegypti*, where mortality was very similar in mud and wood bioassays, and in the susceptible *An. gambiae*, in which mortality was very high on all surfaces. The difference between surface types was least pronounced in *An. gambiae*. Efficacy on mud and cement was not significantly different.

### 3.3. The Effect of Varied Exposure Time on Clothianidin Efficacy

Clothianidin IRS applied to a filter paper killed >80% of exposed susceptible *An. gambiae* Kisumu with an exposure time of 15 min by 120 h post-exposure (Figure 8). When scored at 72 h, mortality was more variable but above 70% in most cases. In contrast, with an exposure time of up to 7 h, the average mortality in the susceptible strain of *Ae. aegypti* (New Orleans) reached 60% only with a 7 h exposure, and in the resistant strain (Cayman), the average mortality never exceeded 20%. In all strains, 100% mortality was observed in the positive controls at all time points (data not shown).

## 4. Discussion

Clothianidin is a potent insecticide that has shown promising efficacy against resistant malaria vectors. In a recent lab study screening the efficacy of repurposed chemistries using CDC bottle bioassays, clothianidin was documented to have the lowest discriminating dose (8.07 µg AI/bottle) out of 11 AIs tested [24], indicating its relatively high potency. In both Phase II hut trials and Phase III village trials, SumiShield™ 50WG, a clothianidin based IRS, has shown good efficacy against susceptible and resistant malaria vectors. In hut trials, this efficacy has been shown to last up to 9 months against wild resistant and lab susceptible *An. gambiae* in Benin [25,26], and in village trials, up to 6 months against resistant *An. culicifacies* in India [9,11]. These results are all based on scoring mortality after a 120 h holding period post-exposure, rather than the standard 24 h [20], based on observations of the delayed mortality and slower acting nature of clothianidin than pyrethroids. Similarly, this study found that mortality exceeded 80% and started to plateau at 72 h post-exposure in the preliminary experiment, and so used 72 h mortality as the endpoint by which to judge the residual efficacy of the IRS.

These studies all observed good residuality of a clothianidin-based IRS. However, differing methodologies, and other uncontrollable variables between these studies, preclude the assessment of efficacy across species, strains, or test surfaces. The completeness of our current study allows us to directly evaluate the efficacy of an IRS formulation against different disease vectors (*An. gambiae*, *An. funestus*, *Ae*. *aegypti*) and nuisance biters (*Cx. quinquefasciatus*), resistant and susceptible strains, and different surface types, which is difficult to achieve in field conditions. The controlled nature of this lab study allows us to evaluate efficacy over time under stable conditions. This allows for direct comparisons between different groups; however, it is important to acknowledge that in real life, conditions are more variable, and therefore residual efficacy observed in the field may be affected by factors, such as physical contact with or cleaning of walls, accumulation of dirt/dust, and fluctuating climatic conditions.

In the present study, using a 72-h holding period, SumiShield™ 50WG was shown to be effective for 18 months on all surfaces tested against susceptible (Kisumu) and resistant (VK7 2014) *An. gambiae*, killing more than 80% of exposed mosquitoes in all but a few anomalous replicates with the resistant strain. This efficacy against a resistant strain is consistent with previous findings of the absence of cross resistance to clothianidin in field populations of *Anopheles* with multiple resistance mechanisms, measured using diagnostic doses of 150 µg AI/bottle in a CDC bottle bioassay [27] in Western Kenya [28] and 2% *w/v* clothianidin on filter papers in a WHO tube test [22] in sites in 16 African countries [29]. For *An. funestus*, results were more variable in this residual efficacy study, and efficacy varied by surface type, likely due to the bioavailability and therefore uptake being affected by the nature of the surface. However, the clothianidin IRS performed consistently better than a deltamethrin IRS formulation against the insecticide-resistant *An. funestus* strain (FUMOZ-R). Against all resistant strains tested, deltamethrin IRS only reached the 80% efficacy threshold in one species, on one surface, at one time-point (*An. funestus* FUMOZ-R strain, mud surface, 24 h post-treatment). In all other instances, mortality remained <80% even when measured up to 7 days post-exposure. Against the susceptible strains, there was more variability, and clothianidin provided little or no increase in efficacy over deltamethrin against *An. gambiae* or *An. funestus*, particularly on wood. Nonetheless, the product, which is based on a neonicotinoid with a different mode of action, offers an advantage over pyrethroid-based products in areas of high resistance, where resistance management recommendations from the WHO would advise against the use of pyrethroid-based vector control in any case [30]. The persistence of the clothianidin IRS efficacy over 6 months (indeed up to 8 months) against susceptible strains has significant impacts on its operational use and an improvement over other IRS formulations currently on the market, some of which have less than 6 months’ efficacy under field conditions. There is thus no clear evidence from this study for cross-resistance to formulated clothianidin in *An. coluzzi* colonised from field sites in Burkina Faso, though there may be some in *An. funestus* colonised from Mozambique. This would have to be confirmed through metabolic or molecular investigation to understand the mechanisms of any cross-resistance, most likely acting through upregulation of detoxifying enzymes. The potential impact of this level of cross-resistance to clothianidin on the field efficacy of the product would also need to be evaluated.

Comparatively, in *Ae. aegypti* (both susceptible and resistant), the 80% efficacy threshold was never reached on any test surface treated with clothianidin IRS. Limited efficacy was also observed in *Cx. quinquefasciatus*, even when exposure time was increased to up to 7 h against both strains, though high variability between replicates and the single time point tested with this strain makes strong conclusions difficult to draw. Differences in susceptibility between the two genera (*Anopheles* and *Aedes*) have been observed previously to a range of chemistries (authors’ observations). This difference appears to be more than the effect of the species’ size, although size can have an impact on susceptibility [31,32,33]. Differences in susceptibility are potentially related to differences in metabolism between species, though the full explanation warrants further investigation. The intrinsic activity of clothianidin, measured by topical application, was identical in a susceptible *Ae. aegypti* New Orleans strain (Appendix A, Appendix A) compared to a susceptible *An. gambiae* Kisumu strain [24], with almost all treated mosquitoes dying after treatment with all concentrations tested. This suggests that the reason for the lower efficacy may be due to reduced tarsal uptake, possibly related to formulation effects, or thicker tarsi in *Aedes*, rather than directly related to the innate potency of the compound. If this is the case, a longer exposure time might have revealed improved efficacy against *Aedes*, though mortality was only increased to around 60% even after 7 h of exposure to a treated surface in the susceptible New Orleans strain. It is important to establish why efficacy varies between genera, as a lack of efficacy against a subset of species may affect the acceptability of the product and could result in poor uptake in the field, particularly in locations where nuisance biters predominate.

Some variation in mortality was observed in the current assay. Variation documented between experimental replicates conducted on the same day is an artifact of testing only 10 mosquitoes per replicate. Variation in efficacy over time was observed; however, this was not linear (e.g., a reduction in 72-h mortality in resistant *An. gambiae* (VK7 2014) was measured at 3 months; however, efficacy was restored at 5 months, and mortality in resistant *An. funestus* exposed to clothianidin was lower earlier than in later months). This may be due to micro-variations in the rearing conditions and fitness of the mosquito cohorts used at different time points, though no correlation between mosquito mortality and weight was clear between time points (Appendix A, Appendix A). As this non-linear variation was not observed in other species at the same timepoint, it is unlikely to be due to the stability of the clothianidin IRS formulation affecting the bioavailability of the AI.

The WHO guidelines for evaluating IRS adulticides state that a 24-h holding period before scoring mortality should be used in bioassays to judge product efficacy [20]. These standard testing methods were developed to assess fast-acting pyrethroid insecticides and so are not suitable for AIs, such as clothianidin, which have a slower acting MoA. This was demonstrated by the current study, when in a preliminary test, >80% efficacy in susceptible and resistant *An. gambiae* was only achieved by 3 days post-exposure. Subsequently, 72-h mortality was used to determine the product’s efficacy. If 24-h mortality had been selected, the residual efficacy over time would be reduced, as has been observed previously [8]. To our knowledge, the post-exposure effect on mosquito behaviour, prior to mortality, has not been systematically studied, but this information would be important to better understand the effect of clothianidin-based products on disease transmission. When products with a novel MoA are tested, it is vital that preliminary studies consider a wider range of outcomes than rapid knock down or kill. For example, any reductions in a mosquito lifespan, or mortality before it has time to become infectious, will have dramatic impacts on a mosquito’s disease transmission potential. By only measuring immediate (within 24 h) mortality, current tests fail to detect outcomes that could significantly relate to the effectiveness of a product under real-life conditions. The additional kill more than 24 h post-exposure to clothianidin is not a limitation for an IRS application, where efficacy stems from a community effect.

Under controlled conditions, clothianidin IRS showed differential efficacy against the surface types tested. Mortality was lower on wood in comparison to cement and mud. Mud and cement are by far the most commonly used housing materials utilised across Africa, on which it is recommended IRS products be evaluated [30], so the higher efficacy on these surfaces is encouraging. To account for this differential efficacy, IRS spray programmes could document house interior surfaces prior to spray treatment and could potentially factor in respraying at earlier time points in houses with wood interiors. This would add a level of logistical complexity, which would be best managed at a community level, though in many areas of malaria endemicity, wood is not as common a building material as mud and particularly concrete. The properties of wood depend on the tree from which it originates due to factors, such as the coarseness of the grain. Beech was used for this study but is not common in Sub-Saharan Africa, and testing residual efficacy against local woods would better predict the performance of an IRS formulation.

Comparing the results from the previously untested backup surfaces with those from the parallel surfaces, which had been treated and stored in the same way but not used for bioassays, demonstrates that using surfaces for bioassays does not diminish the efficacy of the insecticide-treated surfaces. Any loss in efficacy over time can therefore be attributed to physical or chemical changes in the surfaces and/or the applied insecticide, and not a loss of material from the surfaces during bioassays.

## 5. Conclusions

Clothianidin is a potent insecticide against *Anopheles* vectors of malaria, and here we show that in an IRS formulation (SumiShield^TM^ 50WG), it has residual efficacy against *An. gambiae* and *An. funestus* up to at least 18 months on a variety of representative building materials. Long-lasting killing action was demonstrated against a strain of *An. gambiae*, which is resistant to a range of insecticide classes, and although results were less clear against susceptible and resistant strains of *An. funestus*, the clothianidin IRS was far more effective than a deltamethrin-based IRS comparator. Results suggest that the 24-h holding period used to evaluate the efficacy of IRS products may not be suitable for vector control tools based on clothianidin, and a 72-h holding period gives a more accurate measure of its efficacy. However, even with a longer holding period and a much extended exposure period, clothianidin-treated surfaces were not very effective in killing *Ae. aegypti* or *Cx. quinquefasciatus*. Although effective against mosquitoes that transmit malaria, even in areas of high pyrethroid resistance, consideration should, therefore, be given to managing expectations in its performance against nuisance biters, and to the nature of the wall surfaces in the houses where it is to be sprayed. The great potential of this IRS product against mosquitoes that transmit malaria, even in areas of high pyrethroid resistance, is again demonstrated. At 18 months after treatment of surfaces, 100% efficacy was still observed on the key surfaces of mud and cement in resistant and susceptible strains of *An. gambiae* and *An. funestus.*

## Figures and Tables

**Figure 1 insects-13-00112-f001:**
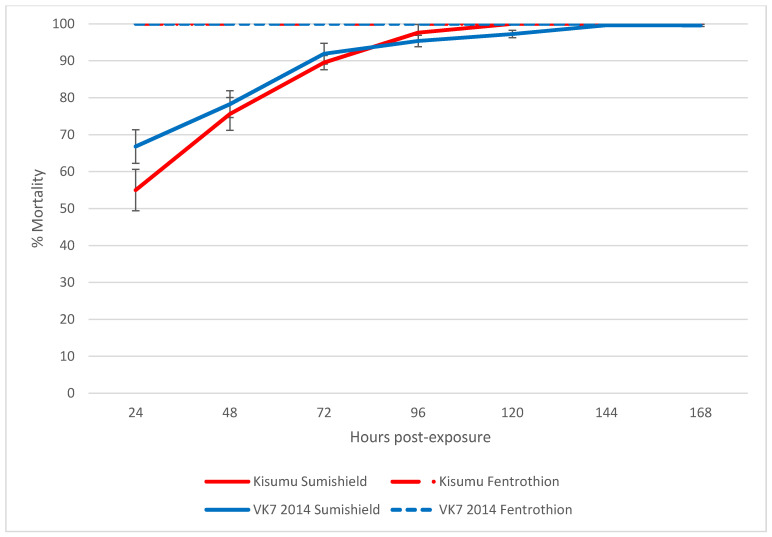
The average cumulative mortality of pyrethroid-susceptible (Kisumu) and -resistant (VK7 2014) strains of *An. gambiae* following exposure to SumiShield^TM^ 50WG or fenitrothion in a WHO tube bioassay. Error bars represent standard error between replicate tubes of ~25 females per tube (*n* = 12 tubes). Abbot’s correction was applied where relevant.

**Figure 2 insects-13-00112-f002:**
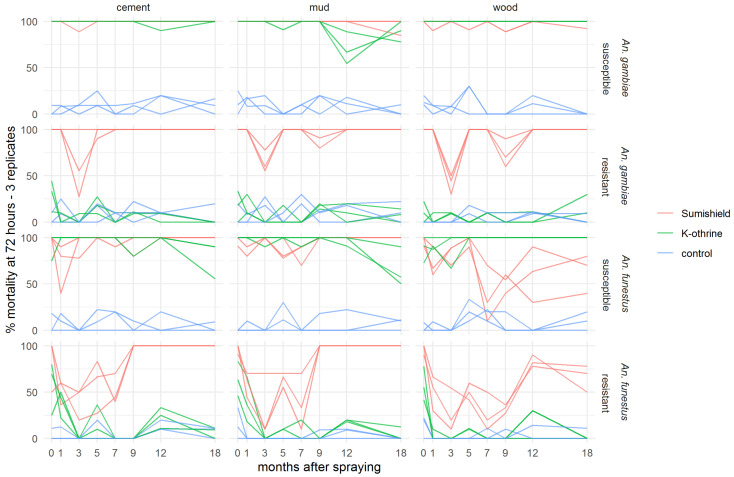
Residual efficacy of K-Othrine^®^ WG250 and SumiShield^TM^ 50WG IRS treatments applied to different surface types against *Anopheles* mosquitoes. Mortality of resistant and susceptible strains of *An. gambiae* and *An. funestus* 72 h after exposure to cement, mud, and wood surfaces treated with clothianidin or deltamethrin IRS is presented, in comparison to control surfaces treated with water only. Mosquitoes were exposed in a WHO cone bioassay at 24 h, and 1, 3, 5, 7, 9, 12, and 18 months after surfaces were treated. Data from 3 replicates of each treatment and surface type are presented as separate lines.

**Figure 3 insects-13-00112-f003:**
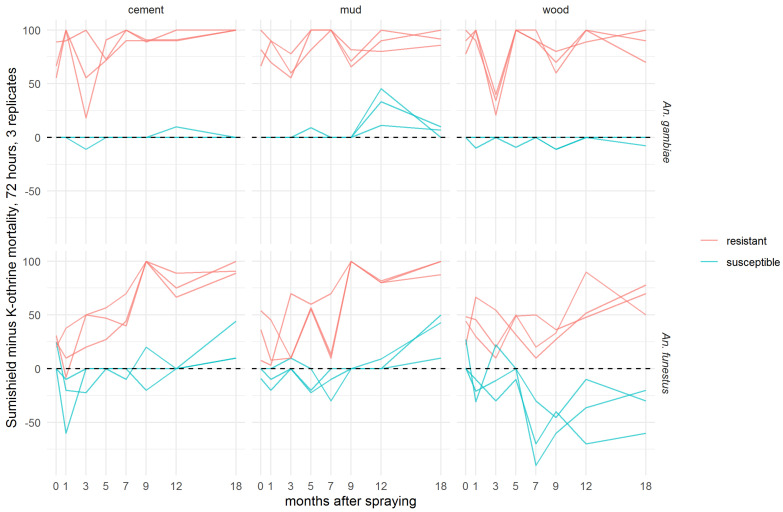
Efficacy of SumiShield^TM^ 50WG compared directly with K-Othrine^®^ WG250 against *Anopheles* mosquitoes. Values represent the additional kill observed over time with clothianidin when testing resistant mosquitoes compared to susceptible mosquitoes. Here, the mortality of mosquitoes from insecticide-susceptible and -resistant strains of *An. gambiae* and *An. funestus* observed 72 h after exposure to cement, mud, or wood surfaces treated with deltamethrin IRS was subtracted from the mortality observed after exposure to clothianidin IRS. Mosquitoes were exposed in a WHO cone bioassay 24 h, 1, 3, 5, 7, 9, 12, and 18 months after treatment. Data from 3 replicates of each treatment and surface type are presented separately.

**Figure 4 insects-13-00112-f004:**
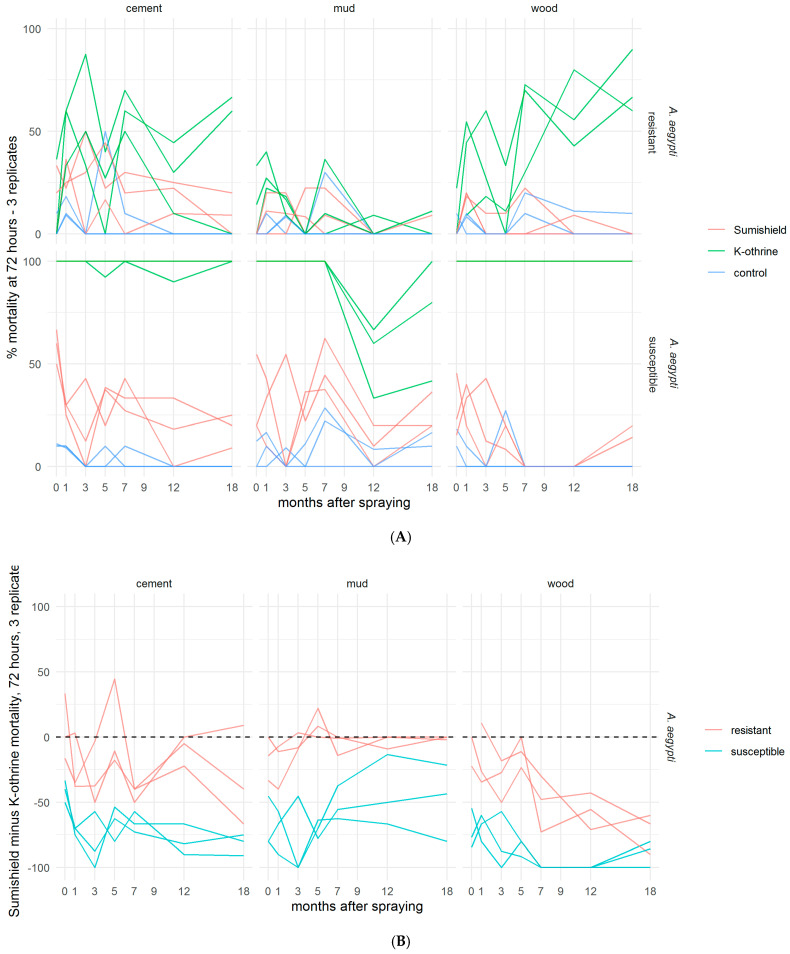
Residual efficacy of K-Othrine^®^ WG250 and SumiShield^TM^ 50WG IRS treatments applied to different surface types against *Aedes aegypti* mosquitoes. (**A**) Mortality of resistant and susceptible strains 72 h after exposure to cement, mud, and wood surfaces treated with clothianidin or deltamethrin IRS, in comparison to control water surfaces. (**B**) Mortality following deltamethrin exposure subtracted from mortality following clothianidin exposure. Values represent the additional kill observed over time with clothianidin when testing resistant mosquitoes compared to susceptible. In both assays (**A**,**B**), mosquitoes were exposed in a WHO cone bioassay 24 h, 1, 3, 5, 7, 9, 12, and 18 months after surfaces were treated. Data from 3 replicates of each treatment and surface type are presented as separate lines.

**Figure 5 insects-13-00112-f005:**
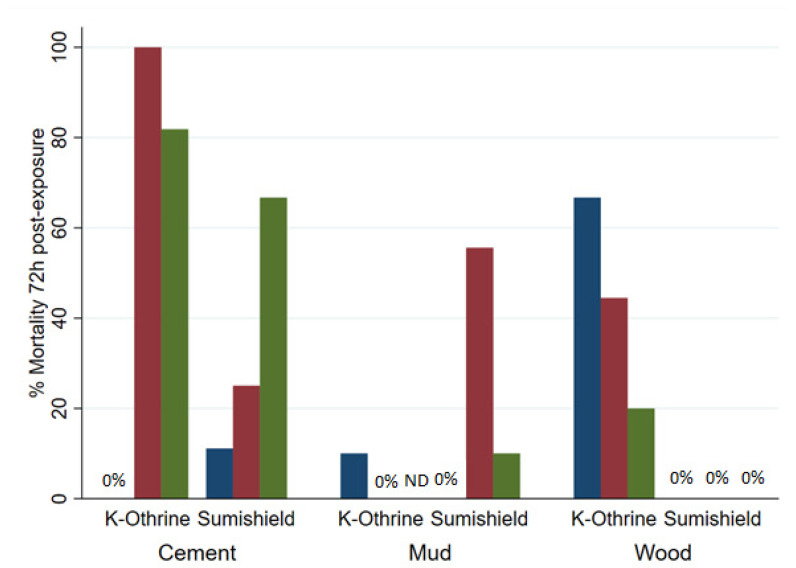
Mortality of *Culex quinquefasciatus* exposed to different surface types treated with K-Othrine^®^ WG250 and SumiShield^TM^ 50WG. Mosquitoes were exposed for 30 min in a WHO cone bioassay 9 months after surfaces were treated. Results represent 3 replicates (blue, red, and green bars) of each surface type, except for deltamethrin-treated mud, where one tile was used to replace a broken tile in the main experiment, and the results of only two replicates are shown.

**Figure 6 insects-13-00112-f006:**
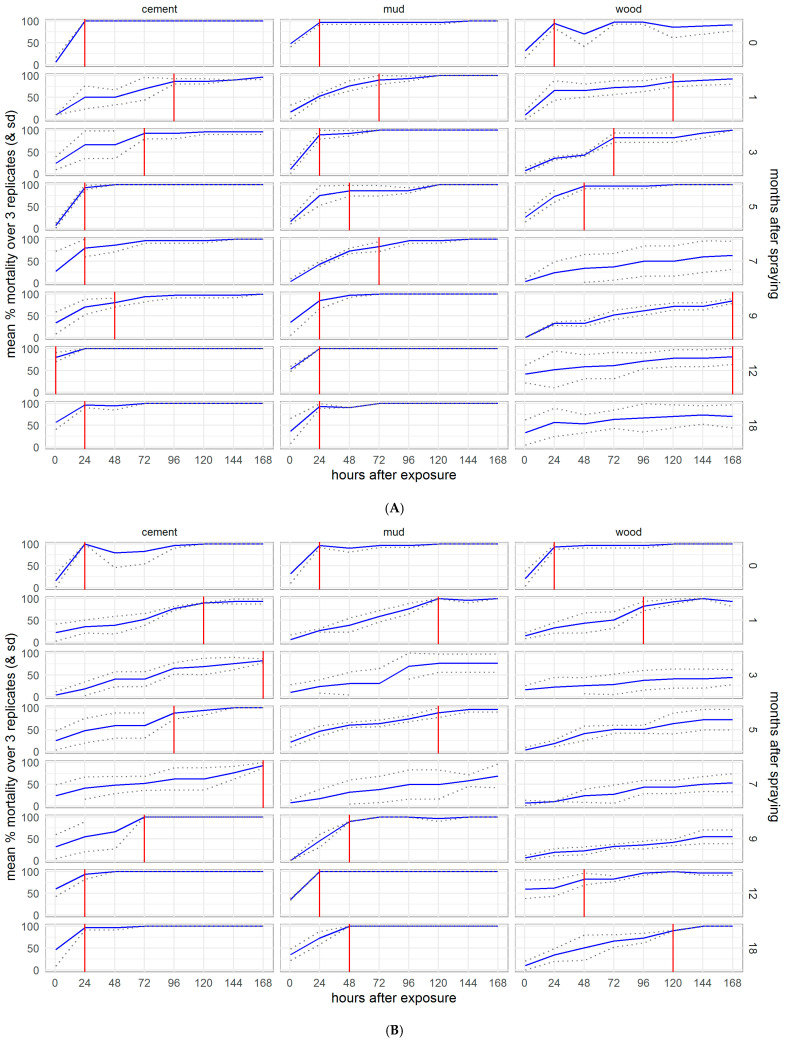
Speed of action of SumiShield^TM^ 50WG against insecticide susceptible (**A**) and resistant (**B**) strains of *Anopheles funestus*. Cumulative mortality is shown by hours post-exposure to treated cement, mud, and wood surfaces, for each month of the experiment. The time to reach the WHO recommended 80% mortality threshold for an IRS product is marked with a red line.

**Figure 7 insects-13-00112-f007:**
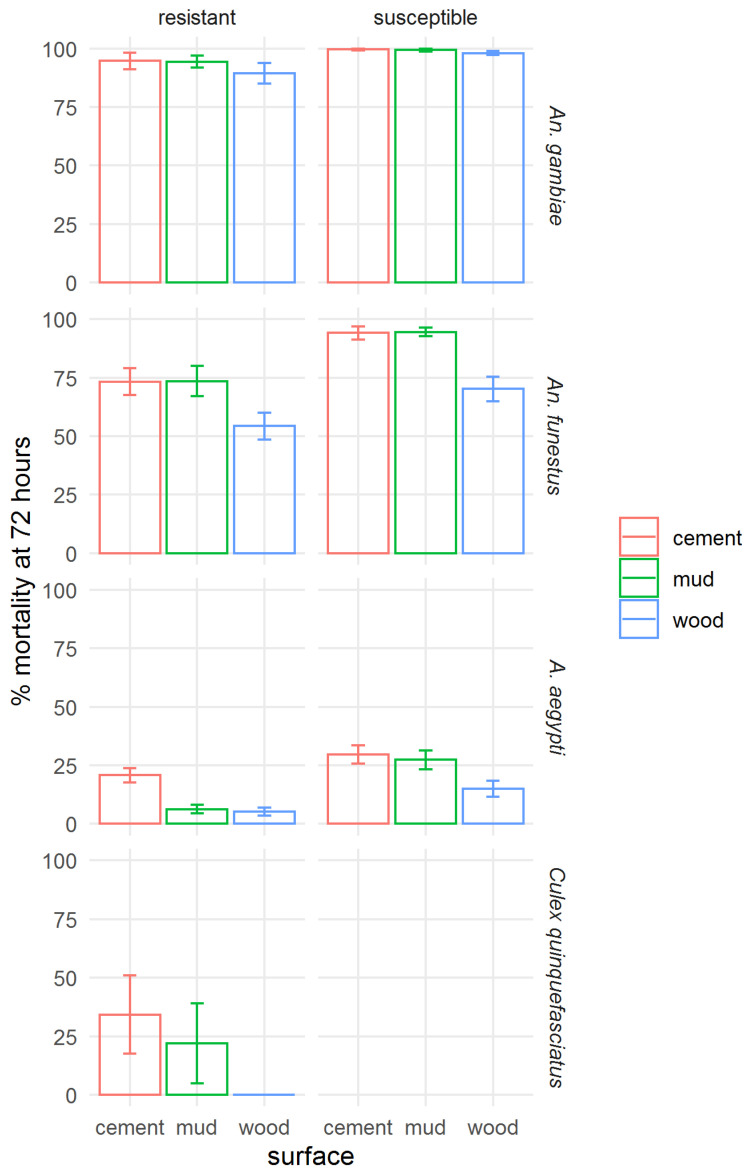
Effect of surface type on the efficacy of SumiShield™ 50WG. Mortality of insecticide susceptible and resistant *Anopheles gambiae*, *An. funestus*, and *Aedes aegypti*, and resistant *Culex quinquefasciatus*. Average 72 h mortality calculated across all replicate bioassays at all time points (0, 1, 3, 5, 7, 9, 12, and 18 months) for each strain and surface type is shown; error bars represent standard error across 3 replicate assays.

**Figure 8 insects-13-00112-f008:**
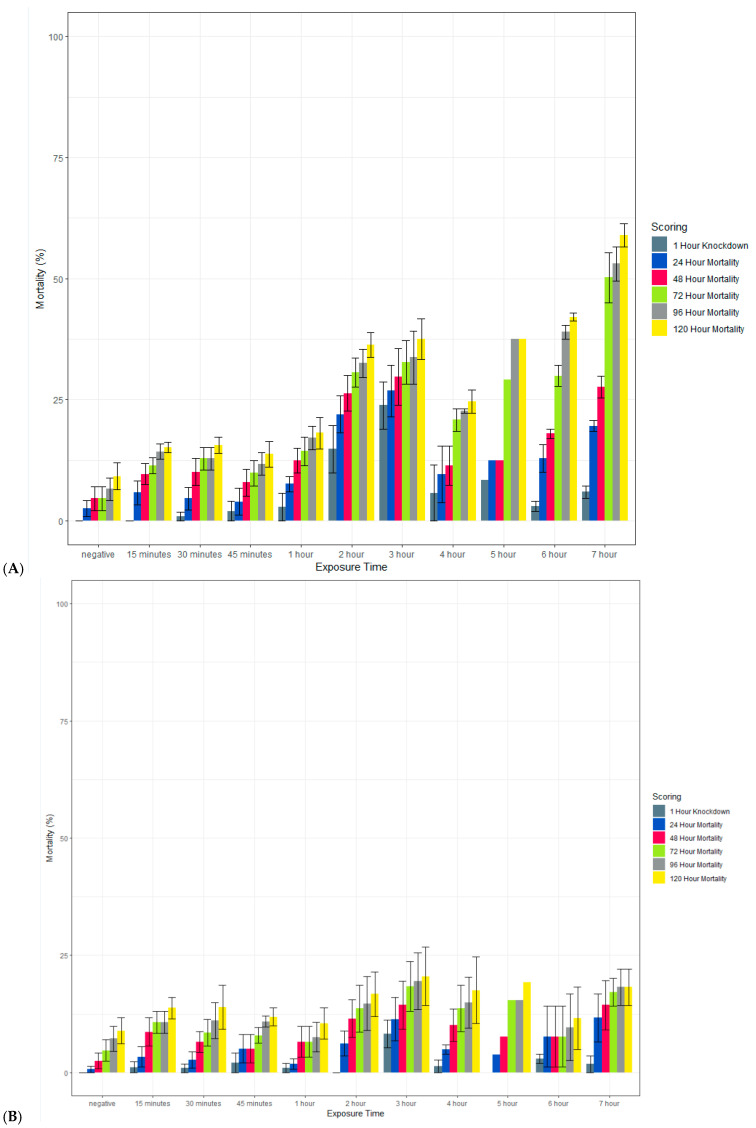
Effect of the length of exposure to SumiShield^TM^ 50WG in a WHO susceptibility tube test against susceptible (New Orleans) (**A**) and resistant (Cayman) (**B**) strains of *Ae. aegypti* and a susceptible strain of *An. gambiae* (Kisumu) (**C**). Mosquitoes were exposed for varying lengths of time to filter papers treated with 13.2 or 733.3 mg AI/m^2^ of clothianidin. Error bars represent standard error between 3 replicate tubes of ~25 females per tube.

## Data Availability

The datasets generated and/or analysed during the current study are not publicly available due to commercial sensitivities but are available from the corresponding author on reasonable request.

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
