# Peer review of "The Residual Efficacy of SumiShield™ 50WG and K-Othrine® WG250 IRS Formulations Applied to Different Building Materials against Anopheles and Aedes Mosquitoes"

_insects, 2022, doi:10.3390/insects13020112_

Round 1
Reviewer 1 Report
This is a well written manuscript that covers the importance question of testing novel insecticides for vector control. The authors cover a range of species, which is excellent, especially as the often-forgotten An. funestus. It is good that the relatively poor effect on nuisance mosquitoes is noted. The issue of potential resistance is also covered.
A few very small editorial notes have been made on the document, and the figure noted must be improved, although as stated, it may be due to the resolution of the figure included.

Author Response
We thank the reviewer for these suggested corrections, which have all been made on the resubmitted manuscript.
Figure 8 has been reformatted, and we hope that it is clearer now. The figure is included as high quality TIFFs as separate files in the submission.
Reviewer 2 Report
The manuscript "The residual efficacy of Sumishield™ 50WG and K-Othrine® WG250 IRS formulations applied to different building materials against Anopheles and Aedes mosquitoes” is a thorough evaluation of the neonicotinoid clothianidin for IRS of mosquitoes, by comparing efficacy against susceptible and resistance mosquito species, multiple substrates, multiple timepoints post-exposure, and multiple mosquito species. To my knowledge, an evaluation with such a wide array of variables has not been previously published.
I found the methodology to be excellent, the results to be of potentially considerable importance to mosquito control, particularly the control of mosquito vectors of malaria, and the conclusions to be justified by the results. I have a few minor comments.
- An overall edit of the manuscript for typographical errors may be helpful. I noticed an unitalicised instance of Anopheles (line 17) and a missing space (line 466), but there may be more.
2. Section 2.1, line 140. It would be helpful if the insecticide resistance of the resistant species could be briefly described in this text, rather sending the reader to another reference just to get a simple description.
3. Section 2.3, lines 190-193. It wasn't clear to me whether the resistant and susceptible strains were exposed to the same surface, i.e. one after the other, or if every treatment was exposed to its own unique substrate. Please clarify. If the susceptible and resistant strains were exposed to the same substrate, which one was exposed first?
4. What is known about the effect of clothianidin on mosquito behaviour during the period between exposure and mortality? Are those mosquitoes able to fly and bite over the two- or three-day interval, or are they moribund? This could affect vectorial capacity.
5. Probably the most confusing result for me is the trend of higher mortality with the length of time post-treatment that the mosquitoes were exposed. Odd that mosquitoes exposed to wood 12 months after spraying would exhibit higher mortality than mosquitoes exposed after 5 months. The authors offer, "There is a trend towards better relative performance of clothianidin in later time points, suggestive of a greater residual efficacy, particularly on mud and cement surfaces". I'm not sure how there could be greater residual, if the surface was only sprayed once? A further discussion of this trend would be useful.
Author Response
We thank the reviewer for their positive comments, and address specific points below.
- An overall edit of the manuscript for typographical errors may be helpful. I noticed an unitalicised instance of Anopheles (line 17) and a missing space (line 466), but there may be more.
We have been through the manuscript carefully and made these corrections and others we found.
- Section 2.1, line 140. It would be helpful if the insecticide resistance of the resistant species could be briefly described in this text, rather sending the reader to another reference just to get a simple description.
Brief description of resistance status of each resistant strain added.
- Section 2.3, lines 190-193. It wasn't clear to me whether the resistant and susceptible strains were exposed to the same surface, i.e. one after the other, or if every treatment was exposed to its own unique substrate. Please clarify. If the susceptible and resistant strains were exposed to the same substrate, which one was exposed first?
The methodology has been clarified and these details added.
- What is known about the effect of clothianidin on mosquito behaviour during the period between exposure and mortality? Are those mosquitoes able to fly and bite over the two- or three-day interval, or are they moribund? This could affect vectorial capacity.
We agree that this is a very important point, but it is not something we studied during these experiments and to the best of our knowledge no such behavioural data has been published for clothianidin. This issue is already addressed in the discussion, but the following sentence has been added to underline the point:
“To our knowledge the post-exposure effect on mosquito behaviour, prior to mortality, has not been systematically studied, but this information would be important to better understand the effect of clothianidin-based products on disease transmission.”
- Probably the most confusing result for me is the trend of higher mortality with the length of time post-treatment that the mosquitoes were exposed. Odd that mosquitoes exposed to wood 12 months after spraying would exhibit higher mortality than mosquitoes exposed after 5 months. The authors offer, "There is a trend towards better relative performance of clothianidin in later time points, suggestive of a greater residual efficacy, particularly on mud and cement surfaces". I'm not sure how there could be greater residual, if the surface was only sprayed once? A further discussion of this trend would be useful.
This particular sentence refers to an increase in performance of clothianidin relative to deltamethrin over time, not an absolute improvement in performance of clothianidin. Although the data is somewhat noisy, Figure 2 would suggest this is a result of declining performance of deltamethrin, rather than an increase in mortality caused by clothianidin, in most cases. The text has been clarified to read:
“There is a trend towards better performance of clothianidin relative to deltamethrin in the later time points, suggestive of a greater residual efficacy of this insecticide, particularly on cement and mud surfaces. In most cases this is a result of declining performance of deltamethrin, though mortality in resistant An. funestus exposed to clothianidin is lower in months 1-7 than in months 9-18 (Figure 2).”
The observation about increasing mortality in resistant An. funestus over time has been added to the section of the Discussion where possible reasons for the noisiness of the bioassay data are discussed:
“Variation in efficacy over time was observed, however was not linear (e.g. a reduction in 72-hour mortality in resistant An. gambiae (VK7 2014) was measured at 3-months, however efficacy was restored at 5-months, and mortality in resistant An. funestus exposed to clothianidin was lower in earlier than in later months).”
This manuscript is a resubmission of an earlier submission. The following is a list of the peer review reports and author responses from that submission.